# SFA-KAN: Spatial-Frequency Aggregation Kolmogorov-Arnold Network for OCT Segmentation

## Abstract

Current medical image segmentation methods exhibit significant limited robustness in optical coherence tomography (OCT) images, primarily attributable to incomplete representation of organ structures and the illumination heterogeneity during image acquisition. To this end, we propose an efficient approach for extracting complete structure and fine-grained details of OCT images, the Spatial-Frequency Aggregation Kolmogorov-Arnold Network (SFA-KAN). Specifically, our method introduces the Spatial-Frequency Aggregation (SFA) module, which operates in the latent space of a convolutional encoder-decoder architecture. This module hierarchically aggregates features from both the spatial and frequency domains. For spatial-domain feature extraction, we propose the Spatial-Shift KAN (S2KA) block, which employs width and height directions channel-mixing KAN linear layers combined with spatial-shift operations. This design facilitates patchwise communication and captures long-distance multi-directional dependencies across the entire image within a single computational pass. For frequency-domain feature extraction, we introduce the Spatial-Shift Frequency Transform (S2FT) block, which employs the same spatial operations as the S2KA block followed by multi-scale fast Fourier transform to isolate clinically-relevant frequency components, enhancing segmentation of anatomically diverse structures. Subsequently, the features from these two different domains are channel-wise concatenated and aggregated via cross attention, enabling the model to reconstruct high-frequency details while preserving global structural integrity. Experiments conducted on two privately collected OCT image datasets employing pixel-based metrics and clinical metrics demonstrated that SFA-KAN achieves state-of-the-art performance for OCT image segmentation. The code will be made publicly available upon acceptance of this paper.

## 1 Introduction

Optical Coherence Tomography (OCT) produces cross-sectional images of both the anterior and posterior segments of the eyes, and the accurate quantitative analysis of universal segmentation of OCT images is crucial for medical research and clinical diagnosis (Qian et al., 2024; Li et al., 2024). However, existing methods for OCT segmentation fail to effectively handle the incomplete representation of organ structures and the illumination heterogeneity during image acquisition, primarily due to the significant variability in imaging conditions, which limits their robustness capability, as shown in Fig. 1.

This paper introduces SFA-KAN, a spatial-frequency aggregation Kolmogorov-Arnold network. While maintaining the encoder-decoder architecture, we innovate by integrating the Spatial-Frequency Aggregation (SFA) module at the bottleneck layer. This module comprises two components: the Spatial-Shift KAN Attention (S2KA) block and the Spatial-Shift Frequency Transform (S2FT) block. The architecture incorporates OCT-specific inductive biases, enhancing robustness against speckle noise and attenuation artifacts.

Our key innovations are:

Figure 1: Motivation: stable and accurate OCT segmentation networks capable of inferring complete structures are significant for ophthalmic diagnosis. (a) The illumination heterogeneity of OCT datasets. (b) The incomplete representation of organ structures.

- We propose the S2KA block, which integrates cyclic width and height directions channel-shifting operations with KAN-linearized projections, enabling bidirectional information flow across multiple spatial directions within a single computational pass and overcoming the limitation of incomplete contextual aggregation in conventional methods.
- We propose the S2FT block, which employs identical spatial-shifting operations followed by multi-scale FFT with adaptive band selectors, to adaptively isolate clinically relevant frequency components and resolve spectral noise and contrast heterogeneity.
- We conduct extensive experiments on two custom-built OCT datasets, which demonstrate our method surpassed the existing methods fully verifying the efficiency of our method.

## 2 RELATED WORKS

Prior works, such as TransUNet (Chen et al., 2024), Swin-UNet (Cao et al., 2022), and UNETR (Hatamizadeh et al., 2022), integrate UNet Ronneberger et al. (2015) with Transformer-based architectures to mitigate the loss of feature representations associated with target structures, which leads to unsmooth object boundaries and compromises accurate identification in OCT images (Wang et al., 2021). The fine segmentation and the structural reasoning of the anterior segment are difficult to achieve simultaneously using Transformer-based methods, not only due to the insufficient multi-scale feature extraction but also because of the heterogeneity caused by data distribution differences in the images. The introduction of transformers has proven effective in reasoning with incomplete imaging data but they neglect the constraints of computational resources in real medical settings. While the state-of-the-art (SOTA) transformer-based medical segmentation model ZigRiR Chen et al. (2025) introduces linear complexity modules for long-distance modeling and significantly enhance computational efficiency, the challenge of substantial intra-class heterogeneity in the data remains unaddressed.

Structured state-space sequence models (SSMs) have demonstrated high efficiency and lightweight performance in long-sequence modeling (Gu & Dao, 2023). Subsequently, VM-UNet Zhang et al. (2024) integrates the visual state-space sequence models into UNet architectures and reduces parameters while maintaining performance comparable to Transformer. However, these models are limited by the linear mapping of SSM equations and lack the capability for nonlinear mapping, a complex nonlinear pattern commonly observed in medical segmentation tasks (Yu et al., 2021), resulting in false-positive issues in segmentation tasks (Wang et al., 2025).

To address the significant heterogeneity across diverse image acquisition scenarios, a more robust capability for nonlinear modeling is required. Therefore, approaches such as UNeXt (Valanarasu & Patel, 2022), Rolling-UNet Liu et al. (2024b) and DPM-Net Wang et al. (2024) combine MLPs with UNet architectures to develop lightweight models that mitigate these challenges. Rolling-UNet and DPM-Net employ distinct spatial transformation strategies on the feature maps extracted by MLPs within the UNeXt framework, effectively capturing long-range dependencies across multiple directions, thereby achieving enhanced performance. The core mechanism of these MLP-based medical segmentation methods is rooted in the MLP-Mixer (Tolstikhin et al., 2021), which incorporates token-mixing MLPs, thereby maintaining robust global modeling capabilities while effectively

reducing computational overhead. However, the global receptive field and spatially-specific architecture limit the generalization capability.

MLPs, particularly when integrated with spatial-shift operations (Wu et al., 2018; Chen et al., 2019), such as $S^2$-MLP (Yu et al., 2022), significantly improve the multi-domain generalization capability of models. Consisting solely of channel-mixing MLP layers without additional token-mixing MLPs, the model is endowed with a localized receptive field, enhancing spatial adaptability and further mitigating model complexity. The Kolmogorov-Arnold Network (KAN) Liu et al. (2024c); Jiang et al. (2025) has emerged as an alternative to MLP-based networks, utilizing learnable activation functions and exhibiting faster convergence to low training loss, all while requiring fewer parameters. This approach demonstrates superior precision and interpretability, particularly in the context of addressing intricate nonlinear relationships. The SOTA KAN-based medical segmentation model MM-UKAN++ (Zhang et al., 2025), along with other UKAN variants (Li et al., 2025), incorporates a multidimensional attention mechanism to weight features from frequency, channel, and spatial perspectives, and has demonstrated superior generalization performance. Moreover, MADGNet Nam et al. (2024) and Y-Net Farshad et al. (2022) further validate that the multi-scale frequency-domain attention mechanisms can effectively capture boundary features and enhance the delineation of tissue contours and anatomical structures, particularly in scenarios requiring fine-grained detail resolution.

Inspired by these works, we propose a novel dual-domain feature aggregation framework designed specifically for high-precision segmentation in clinical ophthalmic imaging.

## 3 METHOD

### 3.1 OVERVIEW

SFA-KAN is illustrated in Fig. 2, which is an encoder-decoder architecture with the SFA module in bottleneck. The SFA module incorporates two core dual-domain feature extraction blocks: the S2KA block and the S2FT block. These features are then concatenated along the channel dimension and aggregated using a cross-attention Liu et al. (2025) mechanism to capture complementary information from both domains. Each encoder block downsamples using max-pooling, and each decoder block up-samples using interpolation. The number of channels across each block is denoted as $C1$ to $C5$. Via the integration of these advanced modules, SFA-KAN substantially enhances robustness and the structural inference capability of OCT structures compared to prior approaches.

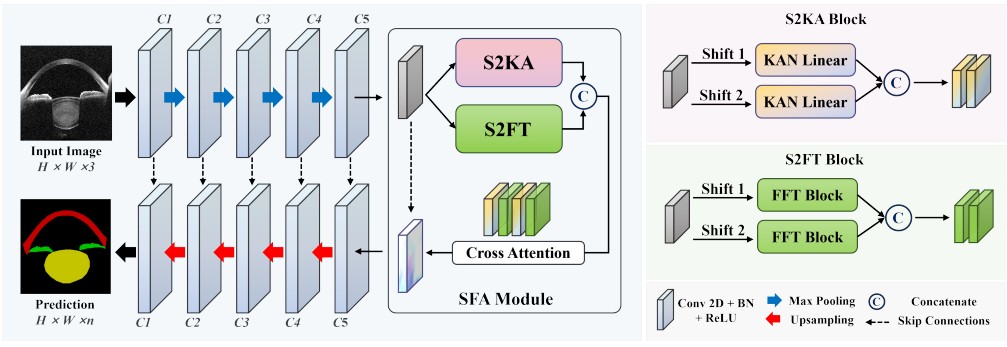

Figure 2: The overview of SFA-KAN, which aggregates dual-domain features via cross-attention in the SFA module. The S2KA and S2FT blocks extract spatial and frequency feature maps, respectively, via KAN Linear and FFT blocks after shift operations in diverse directions.

### 3.2 S2KA BLOCK

As shown in part (a) of Fig. 3, we introduce the S2KA Block based on KAN. The input features are duplicated into two copies, which undergo shift1 and shift2 operations, respectively. Given the input feature matrix $X \in \mathbb{R}^{B \times C \times H \times W}$ with spatial resolution $H \times W$, batch size $B$ and channel number $C$, $X_c$ denotes the $c$-th channel of $X(0 \leq c < C)$. $X_c^W$ and $X_c^H$ denote the cyclic shift operator along

the width and height dimensions, respectively. The compound transformations for the two shifted outputs can be formally expressed as:

$$X_{\text{shift1}} = \left\{ X_c^H(X_c^W(X_c)) \mid X_c \in X \right\}_{c=0}^{C-1} \tag{1}$$

$$X_{\text{shift2}} = \left\{ X_c^H(X_c^W(X_c)) \mid X_c \in X \right\}_{c=0}^{C-1} \tag{2}$$

where the component operations are:

$$X_c^W(X)(h,w) = X(h,(w-c) \mod W) \tag{3}$$

$$X_c^H(X)(h,w) = X((h-c) \mod H, w) \tag{4}$$

where $h \in [0,H)$ and $w \in [0,W)$. To preserve structural integrity, the displaced regions are compensated by cropping and geometrically registering corresponding sections from neighboring feature maps.

After applying the shift operations in both width and height directions, different channels acquire distinct spatial features. The two feature tensors are then fed into KAN Linear for feature transformation, and the outputs are concatenated along the channel dimension. These shifting operations are parameter-free and enable communication between adjacent patches, making long-range contextual interaction feasible. Unlike S2MLP, which adopts unidirectional spatial shifts, the S2KA block introduces diagonal spatial shifts to capture cross-directional dependencies.

In KAN linear layers, instead of learning weights and biases with predefined activation functions, the activation functions themselves are learnable, utilizing gridded basis functions along with trainable scaling factors (Koenig et al., 2024). As shown in part (b) of Fig. 3, a KAN consisting of $K$ layers can be viewed as the interaction between transformation matrices $W$ and activation functions $\sigma$. This process can be formulated as:

$$\text{KAN}(X) = (\Phi_{K-1} \circ \sigma \circ \cdots \circ \Phi_1 \circ \sigma \circ \Phi_0)X \tag{5}$$

where $\Phi_i$ denotes the $i$-th layer of the network and "$\circ$" represents function composition. Each layer has $n_{\text{in}}$ input and $n_{\text{out}}$ output dimensions, and $\Phi$ consists of learnable activation functions $\phi$, which is defined as:

$$\Phi = \{\phi_{q,p}\}, \, p = 1, 2, \ldots, n_{\text{in}}, \, q = 1, 2, \ldots, n_{\text{out}} \tag{6}$$

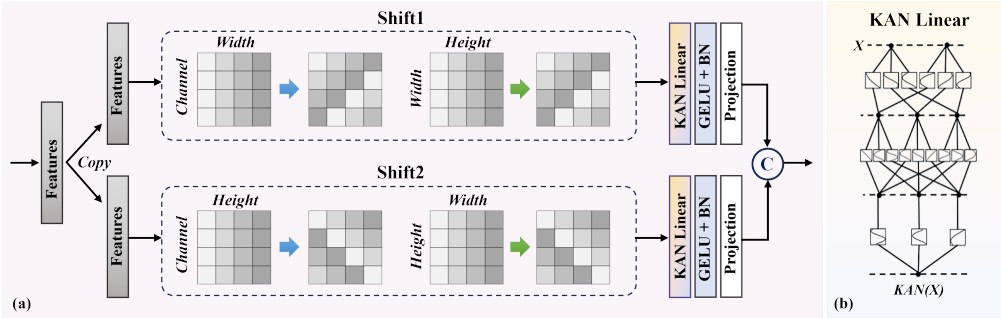

Figure 3: The architecture of S2KA block. (a) Block structure. (b) KAN Layer composition.

The computation from the $k$-th to the $(k+1)$-th layer in the KAN can be represented in matrix form as:

$$\Phi_k = \begin{pmatrix} \phi_{k,1,1}(\cdot) & \phi_{k,1,2}(\cdot) & \cdots & \phi_{k,1,n_k}(\cdot) \\ \phi_{k,2,1}(\cdot) & \phi_{k,2,2}(\cdot) & \cdots & \phi_{k,2,n_k}(\cdot) \\ \vdots & \vdots & \ddots & \vdots \\ \phi_{k,n_{k+1},1}(\cdot) & \phi_{k,n_{k+1},2}(\cdot) & \cdots & \phi_{k,n_{k+1},n_k}(\cdot) \end{pmatrix} \tag{7}$$

where $\phi_{k,p,q}(\cdot)$ denotes the learnable activation function at the position specified by $p$ and $q$ within the $k$-th layer.

### 3.3 S2FT BLOCK

As shown in Fig. 4, the input feature tensor $X \in \mathbb{R}^{B \times C \times H \times W}$ is first duplicated into two copies, which undergo shift1 and shift2 operations, respectively. Subsequently, we introduce a lightweight Fast Fourier Transform (FFT) module to exploit multi-scale frequency-domain information for feature enhancement, comprising multi-scale frequency extraction and dynamic band selection. The two outputs are upsampled then channel-wise concatenated.

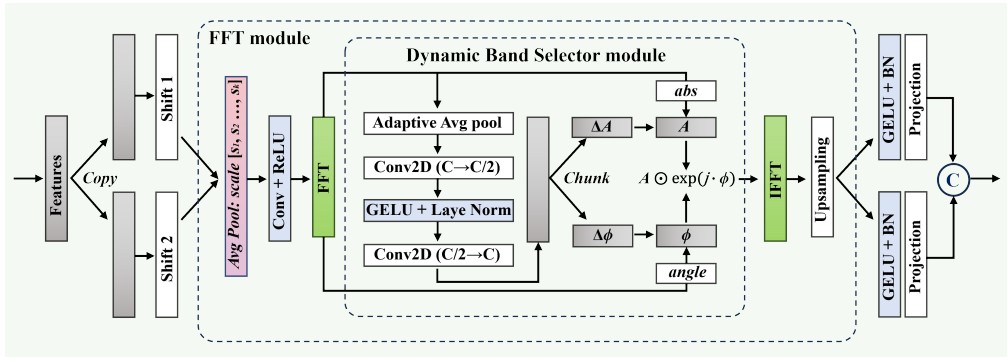

Figure 4: The architecture of S2FT block.

We process the input feature matrix $X$ across $S = \{s_1, s_2, \ldots, s_k\}$ scales to capture frequency components at varying spatial resolution. Specifically, for each scale $s_k$: (1) Spatial downsampling via average pooling with kernel size $s_k$ reduces the feature size to $H/s_k \times W/s_k$, mitigating high-frequency aliasing; (2) The 2D orthogonal fast Fourier transform converts the downsampled features into complex-valued frequency-domain representations. The processed features are then passed to a dynamic band selector module to generate scale-specific parameters to adjust amplitude and phase. An adaptive average pooling reduces spatial dimensions to $1 \times 1$, capturing channel-wise global statistics and a two-layer convolutional bottleneck produces gating parameters $\{\Delta A, \Delta \phi\}$, where $\Delta A \in \mathbb{R}^{B \times C/2 \times 1 \times 1}$ modulates amplitude via $A' = A \cdot (\Delta A + 1)$, and $\Delta \phi \in \mathbb{R}^{B \times C/2 \times 1 \times 1}$ perturbs phase via $\phi' = \phi + 0.1 \cdot \Delta \phi$. Phase perturbations are scaled by 0.1 to maintain stability during learning while allowing gradual phase adjustments. This mechanism avoids manual frequency band tuning, allowing the block to learn optimal frequency transformations end-to-end. The Inverse Fast Fourier Transform (IFFT) is performed by combining the modulated amplitude and phase as:

$$X' = \text{IFFT}(A \odot \exp(j \cdot \phi)) \tag{8}$$

where $j$ represents imaginary unit and $\odot$ represents element-wise multiplication.

## 4 EXPERIMENTS

### 4.1 DATASETS AND IMPLEMENTATION

Currently, publicly available anterior segment OCT datasets remain scarce. Existing open anterior segment datasets are primarily designed for segmenting the cornea and iris under conditions of

Table 1: Overview of OCT datasets.

| Datasets | Pictures | Resolution | Augmented |
|----------|----------|------------|-----------|
| Dataset 1 | 266 | $2135 \times 1468$ | - |
| Dataset 2 | 1330 | $2135 \times 1468$ | Yes |

complete imaging. However, the core motivation of our study lies in addressing imaging defects encountered in practical clinical scans-particularly structural deficiencies of the lens and cornea, as well as data heterogeneity induced by varying scanning conditions. Consequently, current public anterior segment OCT datasets fail to validate the performance of algorithms in these critical aspects.

Furthermore, public posterior segment OCT datasets (i.e., retinal OCT images) are fundamentally distinct from the problem delineated in the motivation section (illustrated in Fig. 1), thus falling outside the scope of our research objectives. Additionally, other vascular and fundus-related datasets categorized as optical coherence tomography angiography (OCTA) images do not constitute conventional OCT imaging in the medical field. They represent distinct ophthalmic imaging modalities that are clinically parallel to OCT; therefore, validating our method using posterior segment data lacks methodological rigor.

To validate the effectiveness of our method, we utilize two anterior segment OCT datasets that we collected as shown in Table 1. Dataset 1 was collected following the identical protocol described in Reference Sun et al. (2024), using the Cirrus HD-OCT 5000 system. The key parameters are as follows: Scan range: 9.0 mm (lateral) $\times$ 2.0 mm (depth), designed to ensure comprehensive coverage from the corneal epithelium to the iris root. Scan speed: 27,000 A-scans per second (Cirrus 5000). Beam Scanning Geometry Correction was activated to compensate for artifacts induced by corneal refraction.

We meticulously selected from clinical data of 50 healthy patients who underwent ophthalmic examinations at our collaborating hospital, with 100 images per patient. Specifically, the dataset was curated from 5,000 images featuring diverse imaging characteristics, ensuring its representativeness and comprehensive reflection of various practical clinical scenarios. All images were annotated by five ophthalmologists with over three years of clinical experience from the collaborating ophthalmic hospital using the LabelMe tool. Given the labor-intensive nature of manual annotation, a total of 266 representative OCT images (resolution: $2135 \times 1468$ pixels) were annotated. Dataset 2 contains 1,330 OCT images derived by augmenting Dataset 1 through random 15-degree rotations and horizontal flipping, aimed at further enhance dataset complexity. This augmentation strategy aligns with real-world clinical conditions, enabling robust validation of the generalization ability of the proposed method.

We resized all the images to $256 \times 256$ and, in accordance with prior research (Guo et al., 2021; Liu et al., 2022), randomly partitioned the datasets into training, validation and testing sets in a $7 : 1 : 2$ ratio. SFA-KAN is implemented using the Pytorch framework (Paszke et al., 2019), with channel numbers set to $C1 = 32$, $C2 = 64$, $C3 = 128$, $C4 = 256$ and $C5 = 512$. The AdamW optimizer is used with a learning rate of $1e - 3$ and the CosineAnnealingLR is employed as the scheduler with a minimum learning rate of $1e - 5$ and a maximum of 50 iterations. We train SFA-KAN for a total of 200 epochs with a batch size of 16. All experiments are performed on a single NVIDIA TITAN RTX GPU.

To evaluate our method, we employ pixel-based metrics including Intersection over Union (IoU), Dice Similarity Coefficient (DSC), Accuracy (Acc), and Average Symmetric Surface Distance (ASSD), and quantitative clinical metrics (Mean Absolute Error, MAE) for clinical measurements comprising Central Corneal Thickness (CCT), mean Iris Thickness (IT), and Lens Thickness (LT). All experiments were conducted with five repeated trials, with results reported as mean ś standard deviation. The calculation formulas for CCT, IT, and LT are as follows:

$$\text{CCT} = r \times |y_{\text{top}} - y_{\text{bot}}| \tag{9}$$

Table 2: Comparison results on Dataset 1.

| Networks | mIoU (%) | DSC (%) | Acc (%) | ASSD ($\mu$m) | CCT ($\mu$m) | IT ($\mu$m) | LT ($\mu$m) |
|---|---|---|---|---|---|---|---|
| UNet | 79.30 ± 0.61 | 86.69 ± 0.45 | 96.28 ± 0.21 | 50.99 ± 2.55 | 55.84 ± 2.80 | 12.01 ± 0.60 | 228.8 ± 25.0 |
| TransUNet | 78.73 ± 0.83 | 86.35 ± 0.62 | 95.04 ± 0.30 | 46.28 ± 3.31 | 33.96 ± 1.70 | 20.20 ± 1.01 | 199.6 ± 15.0 |
| S2-MLP | 80.62 ± 0.49 | 87.50 ± 0.31 | 96.50 ± 0.15 | 43.29 ± 2.16 | 28.86 ± 1.44 | 15.72 ± 0.79 | 198.8 ± 10.0 |
| UNeXt | 75.68 ± 1.20 | 85.75 ± 1.03 | 96.01 ± 0.37 | 46.66 ± 2.83 | 33.96 ± 1.70 | 15.47 ± 2.27 | 177.6 ± 18.9 |
| DPM-Net | 79.90 ± 0.65 | 87.03 ± 0.57 | 95.49 ± 0.21 | 48.71 ± 2.44 | 48.67 ± 2.43 | 11.72 ± 0.59 | 142.8 ± 12.1 |
| Rolling-UNet | 81.48 ± 0.40 | 88.03 ± 0.33 | 96.61 ± 0.12 | 42.80 ± 2.14 | 33.58 ± 1.68 | 13.16 ± 0.66 | 128.0 ± 6.2 |
| UKAN | 74.25 ± 1.06 | 82.91 ± 0.84 | 95.78 ± 0.28 | 55.43 ± 2.77 | 34.34 ± 1.72 | 30.14 ± 1.51 | 173.2 ± 8.7 |
| MM-UKAN++ | 72.79 ± 1.10 | 82.24 ± 0.90 | 94.97 ± 0.46 | 51.47 ± 3.17 | 36.98 ± 1.85 | 33.60 ± 1.68 | 138.0 ± 6.9 |
| Zig-RiR | 75.29 ± 0.92 | 83.97 ± 0.62 | 95.26 ± 0.34 | 57.27 ± 2.86 | 60.37 ± 3.02 | 21.44 ± 1.07 | 151.6 ± 7.6 |
| MADGNet | 78.92 ± 0.51 | 86.45 ± 0.41 | 95.95 ± 0.20 | 47.30 ± 2.36 | 27.73 ± 1.39 | 14.31 ± 0.72 | 151.2 ± 7.6 |
| Y-Net | 81.94 ± 0.45 | 87.47 ± 0.93 | 96.76 ± 0.82 | 40.61 ± 2.35 | 34.33 ± 0.96 | 13.60 ± 1.14 | 207.6 ± 10.1 |
| **SFA-KAN (Ours)** | **84.02 ± 0.27** | **89.61 ± 0.18** | **97.51 ± 0.08** | **30.11 ± 1.63** | **23.01 ± 1.15** | **8.54 ± 0.56** | **104.4 ± 6.3** |

$$\text{LT} = r \times [y'_{\text{top}} - y'_{\text{bot}}] \tag{10}$$

where the vertical distance between the corneal apex $y_{\text{top}}$ and the corresponding lowermost point $y_{\text{bot}}$ defines CCT for the cornea; for LT, $y'_{\text{top}}$ and $y'_{\text{bot}}$ represent the topmost and bottommost surfaces of the lens, respectively. $r$ denotes the pixel-to-micrometer conversion factor ($1\,\text{pixel} = 20\,\mu\text{m}$).

$$\text{IT} = r \times \frac{1}{N} \sum_{x=0}^{W-1} |M(x, y'_{\text{top}}) - M(x, y'_{\text{bot}})| \tag{11}$$

IT was determined by the mean vertical dimension across all columns $x$ in the binary iris mask $M$, where $N$ denotes the number of valid columns containing iris tissue.

## 4.2 COMPARISON AND ANALYSIS

### 4.2.1 ANALYSIS ON FINE-GRAINED DELINEATION

As shown in Table 2, SFA-KAN achieves SOTA performance across all evaluation metrics on Dataset 1. Specifically, building upon the long-range modeling capability brought by two different shift operations and the global feature extraction of KAN, SFA-KAN achieves a DSC of 89.61%, an mIoU of 84.02%, and an Acc of 97.51%. This represents improvements of 1.58% in DSC, 2.54% in mIoU, and 0.90% in Acc compared to the second-best method, Rolling-UNet. The SFA module effectively integrates fine-grained details from complementary spatial and frequency domains, enabling more precise boundary delineation. This capability translates to a 12.69$\mu$m reduction in ASSD compared to Rolling-UNet. Furthermore, SFA-KAN achieves substantially lower errors in key measurements than other recent SOTA methods, specifically: 23.01$\mu$m (CCT error), 8.54$\mu$m (IT error), and 104.4$\mu$m (LT error). Compared to recent Transformer-based and KAN-based methods such as Zig-RiR and MM-UKAN++, and especially compared to MADGNet and Y-Net, which also incorporate multi-scale frequency domain features, SFA-KAN demonstrates superior performance in terms of pixel-level errors.

### 4.2.2 ANALYSIS ON HETEROGENEITY-ROBUST

As shown in Table 3, SFA-KAN achieves SOTA performance across all metrics on Dataset 2, which consists of OCT images exhibiting more severe illumination non-uniformity and structural variations. Compared with recent Transformer-based, KAN-based, and frequency-domain integrated methods, SFA-KAN exhibits significantly less performance degradation, thereby highlighting its robust generalization capability and outperforming the second-best method, Rolling-UNet, by 1.93%

Table 3: Comparison results on Dataset 2.

| Networks | mIoU (%) | DSC (%) | Acc (%) | ASSD ($\mu$m) | CCT ($\mu$m) | IT ($\mu$m) | LT ($\mu$m) |
|---|---|---|---|---|---|---|---|
| UNet | 62.88 ± 0.60 | 67.88 ± 0.55 | 90.71 ± 0.30 | 104.51 ± 5.20 | 63.81 ± 3.10 | 29.09 ± 1.45 | 219.7 ± 10.9 |
| TransUNet | 60.10 ± 0.85 | 69.11 ± 0.70 | 90.87 ± 0.40 | 112.64 ± 5.63 | 42.03 ± 2.10 | 43.35 ± 2.17 | 195.8 ± 9.7 |
| S2-MLP | 59.86 ± 0.50 | 69.46 ± 0.45 | 91.90 ± 0.25 | 104.57 ± 5.23 | 53.28 ± 2.66 | 36.02 ± 1.80 | 169.9 ± 8.5 |
| UNeXt | 58.04 ± 1.20 | 67.80 ± 1.00 | 91.99 ± 0.35 | 106.18 ± 5.31 | 38.56 ± 1.93 | 41.71 ± 2.59 | 160.7 ± 8.0 |
| DPM-Net | 61.33 ± 0.70 | 70.29 ± 0.60 | 92.34 ± 0.28 | 103.29 ± 5.16 | 38.71 ± 1.94 | 30.38 ± 1.52 | 153.9 ± 7.7 |
| Rolling-UNet | 63.82 ± 0.45 | 71.90 ± 0.40 | 92.43 ± 0.24 | 111.59 ± 5.58 | 49.66 ± 2.48 | 39.53 ± 1.98 | 143.8 ± 7.2 |
| UKAN | 58.15 ± 1.10 | 67.96 ± 0.90 | 91.99 ± 0.35 | 105.24 ± 5.26 | 41.28 ± 2.06 | 52.76 ± 2.64 | 158.3 ± 7.9 |
| MM-UKAN++ | 60.91 ± 0.90 | 70.14 ± 0.80 | 91.70 ± 0.40 | 108.78 ± 5.44 | 29.26 ± 1.46 | 27.03 ± 1.35 | 173.0 ± 8.6 |
| Zig-RiR | 59.68 ± 1.30 | 64.20 ± 1.10 | 91.19 ± 0.55 | 118.27 ± 6.41 | 60.68 ± 3.03 | 44.05 ± 2.20 | 216.3 ± 10.8 |
| MADGNet | 62.31 ± 0.55 | 71.09 ± 0.50 | 92.49 ± 0.23 | 106.25 ± 5.31 | 31.64 ± 1.58 | 35.84 ± 1.79 | 143.9 ± 7.2 |
| Y-Net | 63.99 ± 0.14 | 71.38 ± 1.17 | 92.75 ± 0.23 | 109.79 ± 6.04 | 30.99 ± 0.86 | 16.80 ± 0.55 | 164.7 ± 4.7 |
| **SFA-KAN (Ours)** | **65.75 ± 0.30** | **73.54 ± 0.25** | **92.85 ± 0.18** | **96.04 ± 4.80** | **26.90 ± 1.35** | **19.75 ± 0.99** | **131.9 ± 6.6** |

in mIoU, 1.64% in DSC, and 0.42% in Acc. Notably, SFA-KAN demonstrated minimal degradation in key clinical measurements, with increases limited to 3.89 $\mu$m (CCT error), 8.52 $\mu$m (IT error), and 7.5 $\mu$m (LT error). This contrasts sharply with other methods, which showed substantially larger error increases (ranging from $> 10\,\mu$m to $> 25\,\mu$m). This outcome is attributed to the dual-domain approach, which effectively mitigates the illumination heterogeneity challenge inherent in OCT imaging. The qualitative segmentation results are shown in Fig. 5.

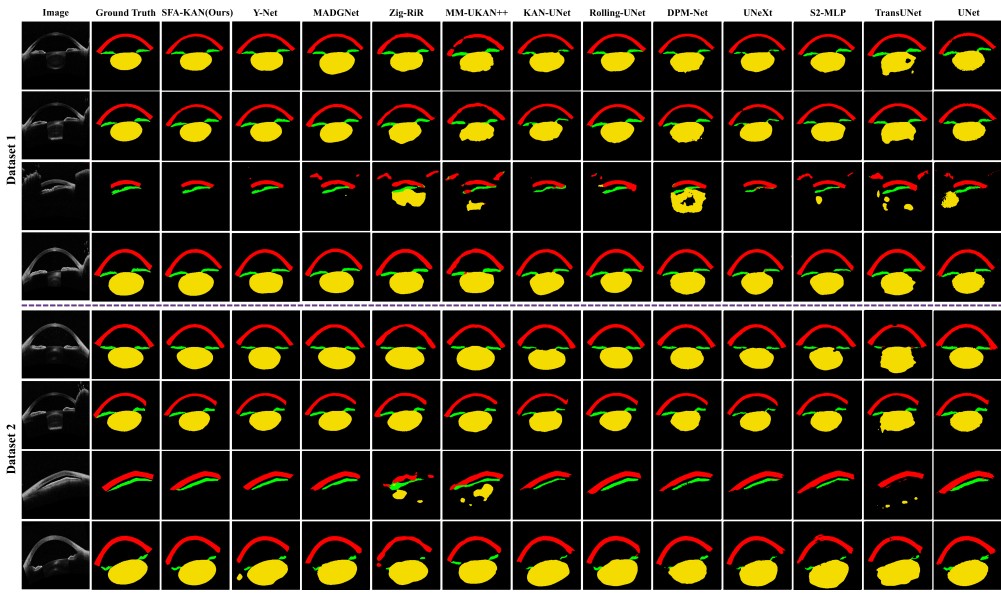

Figure 5: Qualitative comparisons on Dataset 1 and Dataset 2.

### 4.2.3 ANALYSIS ON COMPUTATIONAL COMPLEXITY

We perform a quantitative analysis of computational complexity (encompassing parameter count, GFLOPs, and inference time) to substantiate the efficiency of our methods. As shown in Fig. 6, SFA-KAN outperforms TransUNet significantly in terms of efficiency. With comparable inference speed, it reduces the parameter count and GFLOPs by 64.65% and 53.30%, respectively. Taking the results on Dataset 1 as an example, at an acceptable complexity, our method achieves mIoU improvements of 8.34% and 3.4% compared to lightweight segmentation models UNeXt and S2-

MLP. When compared with KAN-incorporated counterparts UKAN and MM-UKAN++, it delivers substantial mIoU gains of 9.77% and 11.23%, respectively. These results validate that the proposed SFA-KAN achieves a favorable trade-off between segmentation performance and computational efficiency.

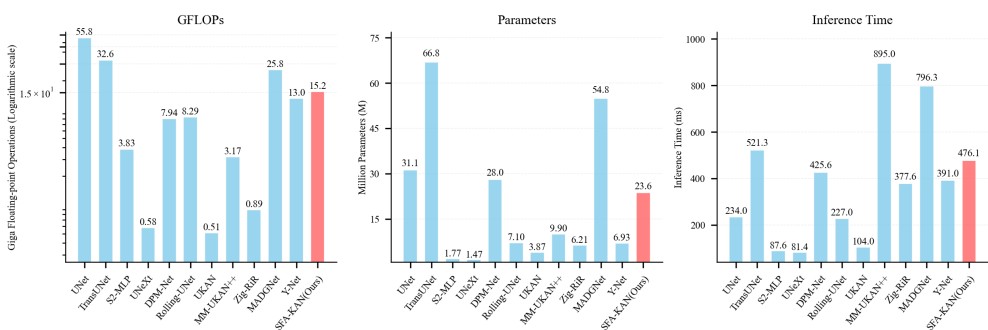

Figure 6: Comparison results of computational complexity.

### 4.2.4 ABLATION RESULTS

We conduct extensive ablation experiments on Dataset 1 to demonstrate the individual contribution of each block in the SFA module of SFA-KAN. As shown in Table 4, we used a symmetric 5-level encoder-decoder architecture with a kernel size of 3 as the baseline. Adding the S2KA block significantly improved pixel-level accuracy, increasing mIoU by 4.17% and DSC by 4.22%, while reducing ASSD by 10.03$\mu$m. This validates its efficacy in global context modeling for fine-grained segmentation. In contrast, the S2FT block optimized on clinical thickness metrics more effectively, increasing mIoU by 3.10% and DSC by 2.59%, while reducing CCT error by 22.66$\mu$m, IT error by 4.07$\mu$m, and LT error by 55.6$\mu$m. The full SFA-KAN architecture demonstrates synergistic superiority: mIoU and DSC reached 83.85% and 89.52%, respectively, with CCT, IT, and LT errors reduced by 14.08$\mu$m, 5.20$\mu$m, and 84.8$\mu$m relative to baseline. The feature visualization of ablation results is shown in Fig. 7.

To further verify the proposed S2KA and S2FT blocks confer distinct advantages over existing spatial- and frequency-capturing modules, comparative experiments were conducted between our domain-specific feature extraction modules and the latest representative methods. The spatial-shift operation enables patch-wise information interaction and captures long-range multi-directional dependencies, thereby facilitating the subsequent extraction of both spatial and frequency domain features, while the KAN layers in the S2KA block demonstrate superior performance relative to methods with weak nonlinear expressiveness such as the Visual State Space (VSS) Liu et al. (2024a) module and the Contrast-Driven Feature Aggregation (CDFA) Lei et al. (2025) module. Methods relying on fixed local window sizes are hampered by low efficiency and prone to the loss of long-range spatial correlations; in contrast, the KAN linear layers in the S2KA block model highly nonlinear spatial correlations (e.g., irregular boundaries and complex textures in medical images) through linear combinations of activation functions, leading to enhanced segmentation effectiveness. Regarding the S2FT block, its dynamic frequency band selection mechanismimplemented post-spatial transformationcan discriminate target frequency domains from other components with higher precision, outperforming the static frequency band partitioning adopted by conventional modules (e.g., Frequency-aware Matching, FAM Bo et al. (2025)). Additionally, although the Multi-Frequency in Multi-Scale Attention (MFMSA) Nam et al. (2024) module attains comparable pixel-level metrics to the S2FT block, its physiological metrics are marginally lower, as the dynamic frequency band selection of the S2FT block enables more accurate localization of segmentation boundaries.

## 5 CONCLUSION

In this paper, we propose a novel architecture, SFA-KAN, for addressing the challenges of illumination heterogeneity and incomplete structure representation in OCT image segmentation. By

Table 4: Ablation study on Dataset 1.

| Networks | mIoU (%) | DSC (%) | Acc (%) | ASSD ($\mu$m) | CCT ($\mu$m) | IT ($\mu$m) | LT ($\mu$m) |
|---|---|---|---|---|---|---|---|
| Baseline | 78.44 ± 0.83 | 84.52 ± 0.72 | 95.95 ± 0.15 | 51.12 ± 3.23 | 48.30 ± 2.09 | 16.43 ± 0.72 | 209.2 ± 10.8 |
| Baseline + VSS | 81.34 ± 0.85 | 88.16 ± 0.60 | 96.05 ± 0.04 | 46.33 ± 6.07 | 41.13 ± 2.38 | 14.61 ± 0.59 | 170.8 ± 6.31 |
| Baseline + CDFA | 81.54 ± 0.92 | 87.47 ± 1.24 | 96.01 ± 0.09 | 45.68 ± 4.45 | 33.32 ± 3.04 | 15.84 ± 0.57 | 178.4 ± 8.3 |
| **Baseline + S2KA** | **82.61 ± 0.44** | **88.74 ± 0.43** | **96.55 ± 0.12** | **41.09 ± 2.25** | **27.09 ± 1.61** | **13.23 ± 0.74** | **161.2 ± 6.0** |
| Baseline + FAM | 80.27 ± 0.92 | 85.53 ± 1.13 | 95.71 ± 0.08 | 41.51 ± 3.66 | 26.03 ± 0.32 | 17.41 ± 0.88 | 187.2 ± 8.4 |
| Baseline + MFMSA | 81.33 ± 0.60 | 86.59 ± 0.59 | 95.66 ± 0.18 | 40.56 ± 0.88 | 38.49 ± 1.87 | 16.88 ± 0.89 | 165.6 ± 5.0 |
| **Baseline + S2FT** | **81.54 ± 0.51** | **87.11 ± 0.64** | **96.10 ± 0.15** | **39.58 ± 2.01** | **25.64 ± 1.08** | **12.36 ± 0.31** | **153.6 ± 4.7** |
| **Baseline + SFA** | **83.85 ± 0.27** | **89.52 ± 0.18** | **97.17 ± 0.08** | **32.63 ± 1.63** | **23.01 ± 1.15** | **11.23 ± 0.56** | **124.4 ± 6.3** |

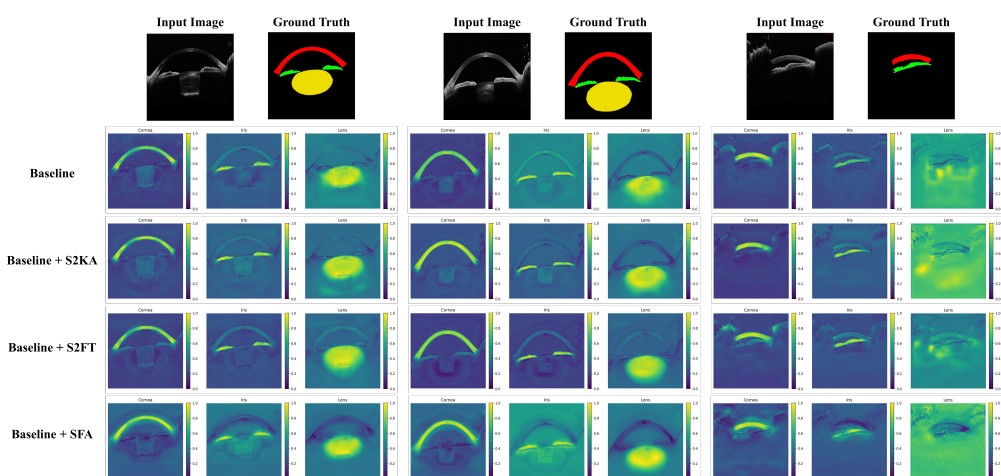

Figure 7: Qualitative comparisons of ablation results on Dataset 1.

integrating the innovative S2KA block for efficient long-range spatial dependency capture and the S2FT block for multi-scale frequency component isolation, the SFA module hierarchically aggregates complementary spatial and frequency features within the latent space. The synergistic cross-attention fusion enables reconstruction of intricate anatomical details while preserving global structural integrity. Extensive validation on two diverse anterior segment OCT datasets demonstrates that SFA-KAN achieves SOTA performance. Quantitative evaluation using both pixel-based metrics and clinically relevant measures confirms its superior accuracy in segmenting critical structures like the cornea, iris, and lens under challenging acquisition variations, providing a reliable foundation for clinical diagnosis via stable, complete organ delineation. Future work will extend SFA-KAN to 3D medical imaging and unsupervised segmentation.

## 6 ETHICS STATEMENT

This retrospective clinical study involving OCT images of the anterior eye segment was approved by the Independent Ethics Committee of a tertiary ophthalmic specialty hospital. The study strictly adhered to the principles of the Declaration of Helsinki. Written informed consent was obtained from all participants prior to data collection, with full disclosure of the study's objectives and data usage scope. All datasets were fully de-identified. The proposed SFA-KAN model is intended solely for clinical decision support and explicitly does not replace the professional judgment of certified ophthalmologists.

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
