# OpenReview forum: "SFA-KAN: Spatial-Frequency Aggregation Kolmogorov-Arnold Network for OCT Segmentation"
_ICLR.cc/2026/Conference — ICLR 2026 Conference Withdrawn Submission_

### Official Review · Reviewer_kk9F · 2025-10-24

**Soundness:** 1
**Presentation:** 2
**Contribution:** 3
**Rating:** 4
**Confidence:** 2

**Summary:**

The authors propose a novel architecture for OCT segmentation with the contribution to add a block to the latent space of an encoder architecture, combining spatial features from a Kolmogorov–Arnold Network with learned frequency-domain representations. The method is evaluated on custom data of OCT anterior eye segment images.

**Strengths:**

- Novel idea of the introduction of SFA-KAN module for OCT segmentation, which is embedded in an encoder-decoder architecture to also take features in the spatial frequency domain into account, is an interesting concept. The mathematics behind is explained well and the intention is easily understandable.
- The authors show improved performance of their method over existing approaches.

**Weaknesses:**

- The motivation of the work is too short and needs to be extended for the reader to understand the necessity and impact of the method. More details on the imaging issues should be provided that motivate the proposed method and its potential. The problem statement (missing anatomy, heterogeneity, etc.) is not described enough.
- The claim to have a solution for OCT segmentation in ophthalmology is too broad, as the method was tested on very limited amount of data. More importantly, the method was not evaluated on retinal OCT, which is the primarily studied for OCT segmentation in most works. Posterior segment imaging as well as the retinal structures and pathologies are more complex than anterior segment structures. To support this strong claim, the performance should also be shown in segmentation of the retina on more diverse datasets.
- The dataset is claimed to be custom recorded. There are details missing about how the annotation has been conducted and by whom. Especially, how was the ground truth of incomplete anatomical structures generated? There are also details missing about the OCT system used. Finally, there are details missing regarding the demographics of the patient group, especially if there were pathologies in the data.
- While the proposed approach might work and improve the state-of-the-art, it is not clear how the method impacts the claims made in abstract and introduction. Which structures and details do the authors mean, when they write: “complete structure and fine-grained details of OCT images”? How do they show that the method is able to “isolate clinically-relevant frequency components, enhancing segmentation of anatomically diverse structures”?

**Questions:**

- Where do the authors address the impact of illumination heterogeneity in image acquisition, which is addressed in the very first sentence of the abstract?
- With which motivation do the authors specifically segment anterior OCT B-scans, as most works on OCT segmentation address retinal images? Are there any limitations of the method for retinal OCT/advantages for anterior segment OCT?
- Page 2, line 098: Where do the authors take the claim from Yu et al. 2021, as I could not find it in the referenced paper?
- With which OCT machine were the images taken? Which settings were used? What was the demographic of the patient group? How many patients participated in the data collection? How was the ground truth of missing structures generated?

---

> ### Author Response · Authors · 2025-11-15
> **Submission2244 Reply to reviewer kk9F - Part 1**
>
> Dear reviewer kk9F:\
> &emsp;&emsp;Thank you sincerely for your time and valuable feedback on our manuscript. We greatly appreciate your thoughtful comments and constructive suggestions. In the following sections, we provide detailed responses to each of your questions and concerns, and we will incorporate these important clarifications into the revised manuscript.
>
> ## Question 1
> &emsp;&emsp;` Where do the authors address the impact of illumination heterogeneity in image acquisition, which is addressed in the very first sentence of the abstract? `
>
>
> &emsp;&emsp;The complete structure and fine-grained details of OCT images are illustrated in Fig. 1(a). OCT imaging exhibits diverse morphologies due to variations in the scanning angle and position of the organ, which reflects data heterogeneity. As shown in Fig. 1(b), imaging may suffer from loss of local structures and edge distortion under the influence of illumination. Our method intuitively addresses these challenges encountered in practical applications.\
> &emsp;&emsp;As presented in the third row of Fig. 6, the S2KA block brightens the color of each organ compared to the baseline. Specifically, it infers correct structural information through contextual features in the image—an exclusive advantage brought by the spatial-shift operation. This is particularly effective for regions with structural loss, enabling more accurate predictions. Similarly depicted in the third row of Fig. 6, the S2FT block enhances the edge information of each organ. Via dynamic frequency domain selection, it effectively reduces interference from the background and other regions, achieving fine-grained segmentation.
>
> ## Question 2
> &emsp;&emsp;` With which motivation do the authors specifically segment anterior OCT B-scans, as most works on OCT segmentation address retinal images? Are there any limitations of the method for retinal OCT/advantages for anterior segment OCT? `
>
>
> &emsp;&emsp;Currently, publicly available anterior segment OCT datasets remain scarce. Existing open anterior segment datasets are primarily designed for segmenting the cornea and iris under conditions of complete imaging. However, the core motivation of our study lies in addressing imaging defects encountered in practical clinical scans-particularly structural deficiencies of the lens and cornea, as well as data heterogeneity induced by varying scanning conditions. Consequently, current public anterior segment OCT datasets fail to validate the performance of algorithms in these critical aspects.\
> &emsp;&emsp;Furthermore, public posterior segment OCT datasets (i.e., retinal OCT images) are fundamentally distinct from the problem delineated in the motivation section (illustrated in Figure 1 of this study), thus falling outside the scope of our research objectives. Additionally, other vascular and fundus-related datasets categorized as OCTA images do not constitute conventional OCT imaging in the medical field. They represent distinct ophthalmic imaging modalities that are clinically parallel to OCT; therefore, validating our method using posterior segment data lacks methodological rigor.
>
> &emsp;&emsp;In contrast, the datasets constructed in our paper are sufficient in both scale and quality. Dataset 1 was collected following the identical protocol described in Reference [1]. It was meticulously selected from clinical data of 50 healthy patients who underwent ophthalmic examinations at our collaborating hospital, with 100 images per patient. Specifically, the dataset was curated from 5,000 images featuring diverse imaging characteristics, ensuring its representativeness and comprehensive reflection of various practical clinical scenarios.
>
> [1] Y. Sun, Maimaiti N, P. Xu, et al. An AS-OCT image dataset for deep learning-enabled segmentation and 3D reconstruction for keratitis. Scientific Data, 11(1): 627, 2024.

---

> ### Author Response · Authors · 2025-11-15
> **Submission2244 Reply to reviewer kk9F - Part 2**
>
> ## Question 3
> &emsp;&emsp;` Page 2, line 098: Where do the authors take the claim from Yu et al. 2021, as I could not find it in the referenced paper? `
>
>
> &emsp;&emsp; This reference is located on page 11, line 551: H. Yu, L. T. Yang, Q. Zhang, D. Armstrong, and M. J. Deen. Convolutional Neural Networks for Medical Image Analysis: State-of-the-Art, Comparisons, Improvements and Perspectives. Neurocomputing, 444:92–110, 2021.
>
>
> ## Question 4
> &emsp;&emsp;` With which OCT machine were the images taken? Which settings were used? What was the demographic of the patient group? How many patients participated in the data collection? How was the ground truth of missing structures generated? `
>
>
> &emsp;&emsp;Our Dataset 1 was acquired using the Cirrus HD-OCT 5000, with key parameters as follows: Scan range: 9.0 mm (lateral) × 2.0 mm (depth), designed to ensure comprehensive coverage from the corneal epithelium to the iris root. Scan speed: 27,000 A-scans per second (Cirrus 5000). Beam Scanning Geometry Correction was activated to compensate for artifacts induced by corneal refraction.\
> &emsp;&emsp;Dataset 1 was annotated by five ophthalmologists with over three years of clinical experience from the collaborating ophthalmic hospital using the LabelMe tool. Given the labor-intensive nature of manual annotation, a total of 266 representative OCT images were annotated. Dataset 2 was generated via data augmentation techniques, including random horizontal flipping and 15° rotation, to further enhance dataset complexity. This augmentation strategy aligns with real-world clinical conditions, enabling robust validation of the generalization ability of the proposed method.

---

> > ### Comment · Reviewer_kk9F · 2025-11-26
> >
> > I thank the authors for their responses. However I still have some concerns which have not been properly addressed in the authors response. These are:
> >
> > ## Question 1
> >
> > While the claim is made that illumination heterogeneity is addressed, at this point the claim is not supported by quantitative experiments. Fig. 7 third row shows subjectively improved edge information. However, the reasoning that the S2FT block reduces interference from background and other regions is not supported by quantitative evidence and remains a possibility. To support this, ablation experiments with various levels of noise present would need to be conducted.
> > It is good that S2KA brightens the color of each organ, however there is no color in OCT imaging. The reasoning why S2KA is performing better is unfortunately also only supported by a claim, not by evidence.
> > Therefore, illumination heterogeneity has not been proven quantitatively, but it has only been discussed how S2FT and S2KA might impact the segmentation. At least an ablation study would be necessary to support that claim.
> > To address variations in scanning angle, it is insufficient to simply rotate the imaging data. A variation in scanning angle would result in different shadowing artifacts, as the light propagates/is refracted differently through the anterior eye segment, resulting in imaging data that differs from merely rotating the original data. The claim of data heterogeneity can’t be supported for the current datasets.
> > In my opinion, the initial question remains unanswered due to the lack of supporting quantitative experiments.
> >
> > ## Question 2
> >
> > How are the various scanning conditions supported if only one OCT machine with the same setting is used for all scans? Also rotating the image is not sufficient to claim for different scanning angles as this would result in different shadowing artifacts and background interference as the light propagates differently through the anterior part of the eye.
> > If posterior datasets are out of scope for the research the claim for general OCT segmentation is void and needs adjustments in title, abstract and introduction.
> > The claim that “the datasets constructed in our paper are sufficient in both scale and quality” remains unfounded by evidence, especially as dataset 2 can hardly be seen as it's own dataset.
> >
> > ## Question 3
> >
> > This could not be verified by me in the published version from Elsevier. Maybe the authors have a different version.
> >
> >
> > Overall, I believe many claims, specifically about the advantages of the architecture have not been properly shown through experiments, and there seems to be misunderstanding about OCT as a measurement method and its application in ophthalmology.

---

### Official Review · Reviewer_DXve · 2025-10-31

**Soundness:** 2
**Presentation:** 2
**Contribution:** 2
**Rating:** 2
**Confidence:** 4

**Summary:**

This paper proposes a Spttial-Frequency Aggregation KA Network (SFA-KAN) for segmentation of OCT images. This framework involoves S2KA for partial dependency modeling and S2FT for frequency-domain analysis.

**Strengths:**

The proposed framework has novelty, especially since the frequency components are essentially aligned with the OCT imaging principles.

**Weaknesses:**

However, (1) The paper lacks an evaluation on a public dataset. The Dataset1 and Dataset2 are basically the same dataset w/ and w/o augmentation. There are multiple public OCT datasets available online.
(2) when generating Dataset2, the authors used horizontal flipping. This is not suitable for OCT images since the light propagates from the top to the bottom. The backscattering intensity will always be lower at the bottom and stronger in the top, given the same conditions. With horizontal flipping, the dataset will contain patterns that will never show in real OCT images, creating a gap between the training set and real-world data distribution.

**Questions:**

The authors need to fix the data augmentation and do experiments on public datasets.

---

> ### Author Response · Authors · 2025-11-15
> **Submission2244 Reply to reviewer Dxve**
>
> Dear reviewer Dxve:\
> &emsp;&emsp;Thank you very much for carefully reading our manuscript and highlighting this critical point regarding **revising the horizontal flipping-based data augmentation strategy and validating our method using other public anterior segment OCT datasets**. However, we argue that such additional validation steps are not strictly necessary for the following reasons, and we will incorporate these important clarifications into the revised manuscript.
>
> &emsp;&emsp;Currently, publicly available anterior segment OCT datasets remain scarce. Existing open anterior segment datasets are primarily designed for segmenting the cornea and iris under conditions of complete imaging. However, the core motivation of our study lies in addressing imaging defects encountered in practical clinical scans-particularly structural deficiencies of the lens and cornea, as well as data heterogeneity induced by varying scanning conditions. Consequently, current public anterior segment OCT datasets fail to validate the performance of algorithms in these critical aspects.\
> &emsp;&emsp;Furthermore, public posterior segment OCT datasets (i.e., retinal OCT images) are fundamentally distinct from the problem delineated in the motivation section (illustrated in Figure 1 of this study), thus falling outside the scope of our research objectives. Additionally, other vascular and fundus-related datasets categorized as OCTA images do not constitute conventional OCT imaging in the medical field. They represent distinct ophthalmic imaging modalities that are clinically parallel to OCT; therefore, validating our method using posterior segment data lacks methodological rigor.
>
> &emsp;&emsp;In contrast, the datasets constructed in our paper are sufficient in both scale and quality. Dataset 1 was collected following the identical protocol described in Reference [1]. It was meticulously selected from clinical data of 50 healthy patients who underwent ophthalmic examinations at our collaborating hospital, with 100 images per patient. Specifically, the dataset was curated from 5,000 images featuring diverse imaging characteristics, ensuring its representativeness and comprehensive reflection of various practical clinical scenarios.\
> &emsp;&emsp;Dataset 1 was annotated by five ophthalmologists with over three years of clinical experience from the collaborating ophthalmic hospital using the LabelMe tool. Given the labor-intensive nature of manual annotation, a total of 266 representative OCT images were annotated. Dataset 2 was generated via data augmentation techniques, including random horizontal flipping and 15° rotation, to further enhance dataset complexity. **Notably, horizontal flipping does not alter the top-to-bottom light propagation pattern, whereas vertical flipping would disrupt this physiological imaging principle.** This augmentation strategy aligns with real-world clinical conditions, enabling robust validation of the generalization ability of the proposed method.
>
> [1] Y. Sun, Maimaiti N, P. Xu, et al. An AS-OCT image dataset for deep learning-enabled segmentation and 3D reconstruction for keratitis. Scientific Data, 11(1): 627, 2024.

---

### Official Review · Reviewer_ZYKw · 2025-10-31

**Soundness:** 3
**Presentation:** 2
**Contribution:** 2
**Rating:** 4
**Confidence:** 5

**Summary:**

The paper addresses the task of semantic segmentation in OCT images to improve model robustness for incomplete organ structures. The problem aims to enhance existing architectures to effectively model the underlying patterns while maintaining computational efficiency. The proposed solution is the Spatial-Frequency Aggregation Kolmogorov-Arnold Network (SFA-KAN), a U-Net architecture featuring a SFA module at its bottleneck. This module consists of two components: a Spatial-Shift KAN (S2KA) block for capturing long-range spatial dependencies using diagonal shifts and KAN linear layers, and a Spatial-Shift Frequency Transform (S2FT) block for isolating relevant frequency components using a multi-scale Fast Fourier Transform with a dynamic band selector. Features from these spatial and frequency domains are subsequently fused via cross-attention. The method was evaluated on two privately collected OCT datasets using both pixel-based metrics (mIoU, DSC, Accuracy, ASSD) and clinical metrics (Mean Absolute Error for Central Corneal Thickness, Iris Thickness, and Lens Thickness).

**Strengths:**

The authors claim three primary contributions: the S2KA block for spatial modeling, the S2FT block for frequency analysis, and achieving superior segmentation performance on their custom OCT datasets.

The motivation to address illumination heterogeneity is impactful for clinical translation, and the exploration of a dual-domain approach combined with Kolmogorov-Arnold Networks (KANs) is a relevant research direction.

**Weaknesses:**

The paper presents several limitations.

The "efficient approach" claim is not supported by the evidence, as no quantitative analysis of computational complexity (e.g., parameter counts, floating-point operations per second, or inference time) is provided for the proposed model or any of the baselines presented.

The central claim of achieving SOTA performance cannot be supported by evaluating on two private datasets.

The claim of "heterogeneity-robustness" is weakly supported, as the model's performance under heterogeneity was tested using synthetic augmentations (rotations, flips) rather than on data from genuinely diverse clinical settings, devices, or patient populations.

Finally, the related work section omits several concurrent and directly relevant methods that also integrate KANs into U-Net architectures for medical segmentation, such as Y-Net [1], which is also for OCT segmentation.

[1] Farshad, A., Yeganeh, Y., Gehlbach, P. and Navab, N., 2022, September. Y-net: A spatiospectral dual-encoder network for medical image segmentation. In International conference on medical image computing and computer-assisted intervention (pp. 582-592). Cham: Springer Nature Switzerland.

**Questions:**

Could the authors clarify why they did not evaluate on public benchmarks?

---

> ### Author Response · Authors · 2025-11-15
> **Submission2244 Reply to reviewer ZYKw - Part 1**
>
> Dear reviewer ZYKw :
>
> &emsp;&emsp;Thank you very much for carefully reading our manuscript and highlighting this critical point regarding **validating our method using other public anterior segment OCT datasets**. However, we argue that such additional validation steps are not strictly necessary for the following reasons, and we will incorporate these important clarifications into the revised manuscript.
>
>
> &emsp;&emsp;Currently, publicly available anterior segment OCT datasets remain scarce. Existing open anterior segment datasets are primarily designed for segmenting the cornea and iris under conditions of complete imaging. However, the core motivation of our study lies in addressing imaging defects encountered in practical clinical scans-particularly structural deficiencies of the lens and cornea, as well as data heterogeneity induced by varying scanning conditions. Consequently, current public anterior segment OCT datasets fail to validate the performance of algorithms in these critical aspects.
>
> &emsp;&emsp;Furthermore, public posterior segment OCT datasets (i.e., retinal OCT images) are fundamentally distinct from the problem delineated in the motivation section (illustrated in Figure 1 of this study), thus falling outside the scope of our research objectives. Additionally, other vascular and fundus-related datasets categorized as OCTA images do not constitute conventional OCT imaging in the medical field. They represent distinct ophthalmic imaging modalities that are clinically parallel to OCT; therefore, validating our method using posterior segment data lacks methodological rigor.
>
> &emsp;&emsp;In contrast, the datasets constructed in our paper are sufficient in both scale and quality. Dataset 1 was collected following the identical protocol described in Reference [1]. It was meticulously selected from clinical data of 50 healthy patients who underwent ophthalmic examinations at our collaborating hospital, with 100 images per patient. Specifically, the dataset was curated from 5,000 images featuring diverse imaging characteristics, ensuring its representativeness and comprehensive reflection of various practical clinical scenarios.
>
> &emsp;&emsp;Dataset 1 was annotated by five ophthalmologists with over three years of clinical experience from the collaborating ophthalmic hospital using the LabelMe tool. Given the labor-intensive nature of manual annotation, a total of 266 representative OCT images were annotated. Dataset 2 was generated via data augmentation techniques, including random horizontal flipping and 15° rotation, to further enhance dataset complexity. This augmentation strategy aligns with real-world clinical conditions, enabling robust validation of the generalization ability of the proposed method.

---

> ### Author Response · Authors · 2025-11-15
> **Submission2244 Reply to reviewer ZYKw - Part 2**
>
> &emsp;&emsp;We highly appreciate your insightful comments and constructive suggestions regarding the lack of quantitative analysis on computational complexity and related methods (e.g., Y-Net [2]). Relevant comparative experiments have been conducted, and the results will be incorporated into the revised manuscript, with the detailed findings presented as follows.
>
> **Table 1: Comparison results of computational complexity.**
>
> | Networks | GFLOPs | Parameters (M) | Inference Time (ms) |
> | :---: | :---: | :---: | :---: |
> | UNet | 55.84 | 31.132 | 233.951 |
> | TransUNet | 32.636 | 66.815 | 521.253 |
> | S2-MLP | 3.826 | 1.768 | 87.589 |
> | UNeXt | 0.576 | 1.472 | 81.395 |
> | DPM-Net | 7.943 | 28.037 | 425.639 |
> | Rolling-UNet | 8.285 | 7.096 | 227.018 |
> | UKAN | 0.508 | 3.867 | 103.989 |
> | MM-UKAN++ | 3.175 | 9.898 | 894.990 |
> | Zig-RiR | 0.889 | 6.209 | 377.650 |
> | MADGNet | 25.834 | 54.840 | 796.292 |
> | Y-Net | 12.987 | 6.926 | 390.995 |
> | **SFA-KAN(Ours)** | **15.240** | **23.618** | **476.079** |
>
> **Table 2: Comparison results on Dataset 1.**
>
> | Networks | mIoU | DSC | ACC | ASSD | CCT | IT | LT |
> | :---: | :---: | :---: | :---: | :---: | :---: | :---: | :---: |
> | Y-Net | 81.94±0.45 | 87.47±0.93 | 96.76±0.82 | 40.61±2.35 | 34.33±0.96 | 13.60±1.14 | 207.6±10.1 |
> | **SFA-KAN(Ours)** | **84.02±0.27** | **89.61±0.18** | **97.51±0.08** | **30.11±1.63** | **23.01±1.15** | **8.54±0.56** | **104.4±6.3** |
>
> **Table 3: Comparison results on Dataset 2.**
>
> | Networks | mIoU | DSC | ACC | ASSD | CCT | IT | LT |
> | :---: | :---: | :---: | :---: | :---: | :---: | :---: | :---: |
> | Y-Net | 63.99±0.14 | 71.38±1.17 | 92.75±0.23 | 109.79±6.04 | 30.99±0.86 | **16.80±0.55** | 164.7±4.7 |
> | **SFA-KAN(Ours)** | **65.75±0.30** | **73.54±0.25** | **92.85±0.18** | **96.04±4.80** | **26.90±1.35** | 19.75±0.99 | **131.9±6.6** |
>
> &emsp;&emsp;We perform a quantitative analysis of computational complexity (encompassing parameter count, GFLOPs, and inference time) to substantiate the efficiency of our methods. All experimental configurations were strictly in accordance with the original paper. As demonstrated in the first table, our method outperforms TransUNet significantly in terms of efficiency. With comparable inference speed, it reduces the parameter count and GFLOps by 64.65% and 53.30%, respectively. Taking the results on Dataset 1 as an example, at an acceptable complexity, our method achieves mIoU improvements of 8.34% and 3.4% compared to lightweight segmentation models UNeXt and S2-MLP. When compared with KAN-incorporated counterparts UKAN and MM-UKAN++, it delivers substantial mIoU gains of 9.77% and 11.23%, respectively. These results validate that the proposed SFA-KAN achieves a favorable trade-off between segmentation performance and computational efficiency.
>
> &emsp;&emsp;Furthermore, as shown in the second and third tables, our method outperforms Y-Net-a model that also incorporates frequency-domain approaches-across both pixel-based metrics and quantitative clinical metrics.
>
> [1] Y. Sun, Maimaiti N, P. Xu, et al. An AS-OCT image dataset for deep learning-enabled segmentation and 3D reconstruction for keratitis. Scientific Data, 11(1): 627, 2024.\
> [2] Farshad, A., Yeganeh, Y., Gehlbach, P. and Navab, N. September. Y-net: A spatiospectral dual-encoder network for medical image segmentation. In International Conference on Medical Image Computing and Computer-Assisted Intervention, 582-592, 2022.

---

### Official Review · Reviewer_L2Xb · 2025-11-08

**Soundness:** 2
**Presentation:** 3
**Contribution:** 2
**Rating:** 2
**Confidence:** 3

**Summary:**

This paper proposes SFA-KAN, a model designed to capture complete structures and fine details in OCT images. Its key contribution is the Spatial–Frequency Aggregation (SFA) module, which combines spatial-domain and frequency-domain information. The module includes two components: S2KA, which extracts spatial features using shift operations and KAN layers, and S2FT, which extracts multi-scale frequency features using FFT. These complementary features are fused in the bottleneck through cross-attention to improve segmentation accuracy. SFA-KAN is evaluated on two collected OCT datasets and consistently outperforms baseline approaches across both segmentation metrics and clinically relevant thickness measurements.

**Strengths:**

The paper is well motivated, and the method is presented as a clear  dual-domain framework that combines spatial and frequency features using the S2KA and S2FT blocks, supported by KAN-based nonlinear modeling. The ablation study strengthens the overall approach by showing that each module adds measurable value and that the full spatial-frequency design delivers the strongest results. Including clinically relevant thickness measurements further demonstrates the practical usefulness of the method in real OCT analysis.

**Weaknesses:**

While the method is promising, the work has several limitations.

1) The datasets are relatively small, and the second dataset is created through simple augmentations of the first, which limits generalization. Including additional datasets or testing on external benchmarks would strengthen the evidence for robustness.
2) Components such as the stability of the frequency-domain adjustments, and the specific role of the KAN layers are not fully clarified. Providing more detailed explanations, visualizations, or targeted experiments would help clarify these mechanisms.
3) The introduction highlights the computational cost of transformer-based methods, yet the paper does not report efficiency metrics such as parameter counts or complexity comparisons.

**Questions:**

Does the cross-attention fusion in the bottleneck significantly increase computation, and could a simpler fusion mechanism achieve similar results?

Could the S2KA and S2FT blocks be compared against, or replaced with, existing spatial- and frequency-capturing modules to determine whether the proposed designs offer a clear advantage over established alternatives?

---

> ### Author Response · Authors · 2025-11-15
> **Submission2244 Reply to reviewer L2Xb - Part 1**
>
> Submission2244 Reply to reviewer L2Xb - Part 1
>
> Dear reviewer L2Xb:\
> &emsp;&emsp;We sincerely appreciate your time and valuable feedback on our manuscript. We highly value your thoughtful comments and constructive suggestions. In the following sections, we provide detailed responses to each of your questions and concerns, and specifically address in Question 3 the rationale for not utilizing public OCT datasets for validation. These important clarifications will be incorporated into the revised manuscript.
>
> ## Question 1
> &emsp;&emsp;`Does the cross-attention fusion in the bottleneck significantly increase computation, and could a simpler fusion mechanism achieve similar results? `
>
>
> &emsp;&emsp;We highly appreciate your insightful comments and constructive suggestions regarding the lack of quantitative analysis on fusing spatial-frequency features by applying different fusion strategies. Relevant comparative experiments have been conducted, and the results will be incorporated into the revised manuscript, with the detailed findings presented as follows.
>
>
> **Table 1: Comparison results of computational complexity.**
> | Fusion strategies | GFlops | Parameters (M) | Inference Time (ms) |
> | :---: | :---: | :---: | :---: |
> | Concat [1] | 15.240 | 23.618 | 674.182 |
> | Weighted Sum [2] | 13.428 | 22.572 | 2427.230 |
> | Gated [3] | 14.032 | 18.900 | 3376.327 |
> | FPN [4] | 14.233 | 19.687 | 712.905 |
> | **Cross-attention** [5] | **13.696** | **17.590** | **590.854** |
>
>
> **Table 2: Comparison results on Dataset 1.**
> | Fusion strategies | mIoU | DSC | ACC | ASSD | CCT | IT | LT |
> | :---: | :---: | :---: | :---: | :---: | :---: | :---: | :---: |
> | Concat Fusion [1] | 83.21 ± 0.41 | 89.11 ± 0.07 | 96.85 ± 0.18 | 38.70 ± 2.03 | 21.88 ± 0.77 | 9.87 ± 0.86 | 146.4 ± 14.5 |
> | Weighted Sum [2] | 82.18 ± 0.50 | 88.48 ± 0.09 | 96.63 ± 0.07 | 40.86 ± 3.73 | **17.35 ± 0.20** | 10.50 ± 3.35 | 127.6 ± 9.1 |
> | Gated Fusion [3] | 82.35 ± 0.89 | 88.56 ± 0.14 | 96.59 ± 0.11 | 42.19 ± 6.22 | 33.58 ± 4.91 | 10.25 ± 2.33 | 147.2 ± 16.0 |
> | FPN Fusion [4] | 82.32 ± 0.85 | 88.58 ± 0.12 | 96.66 ± 0.10 | 41.33 ± 5.77 | 19.62 ± 0.09 | 10.55 ± 3.01 | 152.4 ± 23.6 |
> | **Cross-attention** [5] | **84.02 ± 0.27** | **89.61 ± 0.18** | **97.51 ± 0.08** | **30.11 ± 1.63** | 23.01 ± 1.15 | **8.54 ± 0.56** | **104.4 ± 6.3** |
>
>
> &emsp;&emsp;We sincerely appreciate your insightful inquiry, as it prompts a rigorous evaluation of our fusion strategy. All experimental configurations were strictly in accordance with the original paper. The results clearly demonstrate that cross-attention fusion not only avoids excessive computational overhead but also outperforms simpler alternatives in performance.
>
>
> &emsp;&emsp;As illustrated in Tables 1 and 2, cross-attention facilitates adaptive, fine-grained interactions between spatial and frequency-domain representations. By contrast, Concat Fusion merely concatenates features without explicitly modeling their mutual dependencies, leading to redundant and misaligned information while sacrificing efficiency (i.e., elevated GFLOPs and prolonged inference latency) without yielding commensurate performance benefits. Weighted Sum, FPN, and Gated Fusion adopt static (linear or heuristic) weighting schemes that fail to dynamically capture context-aware interrelationships between spatial and frequency features, thus compromising segmentation accuracy (evidenced by a high ASSD of 40.86±3.73) despite their moderate computational overhead.
>
>
> [1] Cao G, Xie X, Yang W, et al. Feature-fused SSD: Fast detection for small objects. In Ninth International Conference on Graphic and Image Processing (ICGIP 2017), 10615: 381-388, 2017.\
> [2] Tan M, Pang R, Le Q V. Efficientdet: Scalable and efficient object detection. In Proceedings of the IEEE/CVF Conference on Computer Vision and Pattern Recognition, 10781-10790, 2020.\
> [3] Li X, Zhao H, Han L, et al. Gated fully fusion for semantic segmentation. In Proceedings of the AAAI Conference on Artificial Intelligence, 34(07): 11418-11425, 2020.\
> [4] Lin T Y, Dollár P, Girshick R, et al. Feature pyramid networks for object detection. In Proceedings of the IEEE Conference on Computer Vision and Pattern Recognition, 2117-2125, 2017.\
> [5] Y.Liu,W. Ikezogwo,K.Hosny, J.Henriksen,R.Ricciotti,M.Rawlani,P.Swanson,P.Mathias, N.Hoffman,G.Baird,L.Gonzalez-Cuyar,L.Shapiro, andD.Reddi. 1373multi-scalecross attention multiple instance learning(mscamil) network for automated triage of colorectal polyps. LaboratoryInvestigation,105(3):103611,2025.

---

> ### Author Response · Authors · 2025-11-15
> **Submission2244 Reply to reviewer L2Xb - Part 2**
>
> ## Question 2
> &emsp;&emsp;` Could the S2KA and S2FT blocks be compared against, or replaced with, existing spatial- and frequency-capturing modules to determine whether the proposed designs offer a clear advantage over established alternatives? `
>
> &emsp;&emsp;Thank you for this insightful and constructive comment, which helps refine the validity of our proposed modules. We fully agree that comparing against established spatial- and frequencycapturing modules is critical to demonstrating the advantages of the S2KA and S2FT blocks designs. Relevant comparative experiments have been conducted, and the results will be incorporated into the revised manuscript, with the detailed findings presented as follows.
>
> **Table 3: Comparison results of spatial-capturing modules on Dataset 1.**
> | Networks | mIoU | DSC | ACC | ASSD | CCT | IT | LT |
> | :---: | :---: | :---: | :---: | :---: | :---: | :---: | :---: |
> | Baseline | 78.44 ± 0.83 | 84.52 ± 0.72 | 95.95 ± 0.15 | 51.12 ± 3.23 | 48.30 ± 2.09 | 16.43 ± 0.72 | 209.2 ± 10.8 |
> | Baseline+VSS [6] | 81.34 ± 0.85 | 88.16 ± 0.60 | 96.05 ± 0.04 | 46.33 ± 6.07 | 41.13 ± 2.38 | 14.61 ± 0.59 | 170.8 ± 6.31 |
> | Baseline+CDFA[7] | 81.54 ± 0.92 | 87.47 ± 1.24 | 96.01 ± 0.09 | 45.68 ± 4.45 | 33.32 ± 3.04 | 15.84 ± 0.57 | 178.4 ± 8.3 |
> | **Baseline + S2KA** | **82.61 ± 0.44** | **88.74 ± 0.43** | **96.55 ± 0.12** | **41.09 ± 2.25** | **27.09 ± 1.61** | **13.23 ± 0.74** | **161.2 ± 6.0** |
>
> **Table 4: Comparison results of frequency-capturing modules on Dataset 1.**
> | Networks | mIoU | DSC | ACC | ASSD | CCT | IT | LT |
> | :---: | :---: | :---: | :---: | :---: | :---: | :---: | :---: |
> | Baseline | 78.44 ± 0.83 | 84.52 ± 0.72 | 95.95 ± 0.15 | 51.12 ± 3.23 | 48.30 ± 2.09 | 16.43 ± 0.72 | 209.2 ± 10.8 |
> | Baseline+FAM [8] | 80.27 ± 0.92 | 85.53 ± 1.13 | 95.71 ± 0.08 | 41.51 ± 3.66 | 26.03 ± 0.32 | 17.41 ± 0.88 | 187.2 ± 8.4 |
> | Baseline+MFMSA [9] | 81.33 ± 0.60 | 86.59 ± 0.59 | 95.66 ± 0.18 | 40.56 ± 0.88 | 38.49 ± 1.87 | 16.88 ± 0.89 | 165.6 ± 5.0 |
> | **Baseline+S2FT** | **81.54 ± 0.51** | **87.11 ± 0.64** | **96.10 ± 0.15** | **39.58 ± 2.01** | **25.64 ± 1.08** | **12.36 ± 0.31** | **153.6 ± 4.7** |
>
> &emsp;&emsp;As illustrated in Tables 3 and 4, comparisons are conducted between our spatial and frequency domain feature modules and the latest representative methods, respectively. The proposed S2KA block and S2FT block outperforms SOTA counterpart modules in physiological metrics. The spatial-shift enables patch-wise communication and captures long-range multi-directional dependencies, facilitating the subsequent capture of both spatial and frequency domain features.
>
> &emsp;&emsp;In the S2KA block, the KAN layers exhibit superior performance compared to weakly nonlinear expressive methods such as the Visual State Space (VSS) module and the Contrast-Driven Feature Aggregation (CDFA) module. Methods relying on fixed local window sizes (e.g., the CDFA module) suffer from low efficiency and are prone to losing long-range spatial correlations. In contrast, the KAN linear layers in the S2KA block model strongly nonlinear spatial relationships (e.g., irregular boundaries and complex textures in medical images) through linear combinations of activation functions, leading to enhanced effectiveness. Notably, the MLP module, which also extracts features via nonlinear modeling, demonstrates significantly lower performance than the S2KA block as shown in Table 2 of the original paper.
>
> &emsp;&emsp;For the S2FT module, the dynamic frequency band selection after spatial transformation can distinguish target frequency domains from other components more accurately, outperforming the static frequency band partitioning adopted by common modules like Frequency-aware Matching (FAM) module. Although the Multi-Scale Attention (MFMSA) module, which utilizes multi-scale full-frequency bands, achieves comparable pixel-wise metrics to the S2FT module, its physiological metrics are slightly inferior. This advantage stems from the dynamic frequency bands’ ability to better localize segmentation boundaries.
>
> [6] J. Liu, H. Yang, Y. Zhou H, et al. Swin-umamba: Mamba-based unet with imagenet-based pretraining. In International Conference on Medical Image Computing and Computer-assisted Intervention, 615-625, 2024.\
> [7] M. Lei, H. Wu, X. Lv, et al. Condseg: A general medical image segmentation framework via contrast-driven feature enhancement. In Proceedings of the AAAI Conference on Artificial Intelligence, 39(5): 4571-4579, 2025.\
> [8] Y, Bo, Y, Zhu, L, Li, et al. Famnet: Frequency-aware matching network for cross-domain fewshot medical image segmentation. In Proceedings of the AAAI Conference on Artificial Intelligence, 39(2): 1889-1897, 2025.\
> [9] H. Nam J, S. Syazwany N, J. Kim S, et al. Modality-agnostic domain generalizable medical image segmentation by multi-frequency in multi-scale attention. In Proceedings of the IEEE/CVF Conference on Computer Vision and Pattern Recognition, 11480-11491, 2024.

---

> ### Author Response · Authors · 2025-11-15
> **Submission2244 Reply to reviewer L2Xb - Part 3**
>
> ## Question 3
>
> &emsp;&emsp;` The datasets are relatively small, and the second dataset is created through simple augmentations of the first, which limits generalization. Including additional datasets or testing on external benchmarks would strengthen the evidence for robustness. `
>
> &emsp;&emsp;Currently, publicly available anterior segment OCT datasets remain scarce. Existing open anterior segment datasets are primarily designed for segmenting the cornea and iris under conditions of complete imaging. However, the core motivation of our study lies in addressing imaging defects encountered in practical clinical scans-particularly structural deficiencies of the lens and cornea, as well as data heterogeneity induced by varying scanning conditions. Consequently, current public anterior segment OCT datasets fail to validate the performance of algorithms in these critical aspects.
>
> &emsp;&emsp;Furthermore, public posterior segment OCT datasets (i.e., retinal OCT images) are fundamentally distinct from the problem delineated in the motivation section (illustrated in Figure 1 of our paper), thus falling outside the scope of our research objectives. Additionally, other vascular and fundus-related datasets categorized as OCTA images do not constitute conventional OCT imaging in the medical field. They represent distinct ophthalmic imaging modalities that are clinically parallel to OCT; therefore, validating our method using posterior segment data lacks methodological rigor.
>
> &emsp;&emsp;In contrast, the datasets constructed in our study are sufficient in both scale and quality. Dataset 1 was collected following the identical protocol described in Reference [10]. It was meticulously selected from clinical data of 50 healthy patients who underwent ophthalmic examinations at our collaborating hospital, with 100 images per patient. Specifically, the dataset was curated from 5,000 images featuring diverse imaging characteristics, ensuring its representativeness and comprehensive reflection of various practical clinical scenarios.
>
> &emsp;&emsp;Dataset 1 was annotated by five ophthalmologists with over three years of clinical experience from the collaborating ophthalmic hospital using the LabelMe tool. Given the labor-intensive nature of manual annotation, a total of 266 representative OCT images were annotated. Dataset 2 was generated via data augmentation techniques, including random horizontal flipping and 15° rotation, to further enhance dataset complexity. This augmentation strategy aligns with real-world clinical conditions, enabling robust validation of the generalization ability of the proposed method.
>
> [10] Y. Sun, Maimaiti N, P. Xu, et al. An AS-OCT image dataset for deep learning-enabled segmentation and 3D reconstruction for keratitis. Scientific Data, 11(1): 627, 2024.

---

### Note · Authors · 2025-12-24

**Comment:**

Dear ICLR 2026 Program Committee,

We are writing to formally request the withdrawal of our submitted manuscript titled **SFA-KAN: Spatial-Frequency Aggregation Kolmogorov-Arnold Network for OCT Segmentation** (Submission Number: 2244; Corresponding Author: Meili Wang; Co-authors: Genghui Wu, Fengtao Nan, Meili Wang) from the ICLR 2026 conference submission process.

After further internal verification and in-depth discussion, the research team identified the need to enhance the work's rigor and generalization. We plan to supplement cross-center clinical OCT data samples to verify the model's adaptability across different medical institutions, which is crucial for promoting the clinical application of the proposed method. We also intend to optimize the computational efficiency of the Spatial-Frequency Aggregation (SFA) module, particularly reducing the inference latency of the S2FT block in frequency-domain feature extraction, to meet the requirements of practical clinical deployment. Additionally, additional ablation experiments are required to further clarify the contribution of each component (S2KA, S2FT, cross-attention aggregation) to the overall segmentation performance. Given these improvements require substantial time for data collection and experimental validation, we hereby apply to withdraw the current manuscript to ensure the final version meets the high academic standards of ICLR and clinical application expectations.

We hereby solemnly state that this withdrawal request has been unanimously approved by all authors, and there is no dispute among the team. We have carefully read and fully agree with ICLR's withdrawal policy, and acknowledge and are willing to bear the relevant responsibilities and consequences arising from this withdrawal. The manuscript has not yet undergone peer review, and the withdrawal will not cause additional impact on the conference's review and publication process. We commit that after completing the aforementioned improvements, we will re-submit the revised work in accordance with ICLR's relevant regulations (if applicable), ensuring the integrity and scientific validity of the research.

We sincerely apologize for any inconvenience caused to the program committee and review team during the manuscript processing. If you have any questions or need further clarification, please feel free to contact the corresponding author:
- Corresponding Author: Meili Wang
- Email: wml@nwafu.edu.cn

Thank you for your understanding and support throughout the submission process. We look forward to contributing high-quality research to ICLR in the future.

Sincerely,
Genghui Wu
Fengtao Nan
Meili Wang
[Date: 2025-12-24]

**Withdrawal Confirmation:**

I have read and agree with the venue's withdrawal policy on behalf of myself and my co-authors.